# Comparative Study and Multi-Objective Crashworthiness Optimization Design of Foam and Honeycomb-Filled Novel Aluminum Thin-Walled Tubes

**Yi Tao [1], Yonghui Wang [2], Qiang He [2,\*], Daoming Xu [1] and Lizheng Li [2]**

[1]   Marine Design & Research Institute of China, Shanghai 200011, China
[2]   School of Mechanical Engineering, Jiangsu University of Science and Technology, Zhenjiang 212000, China
\*    Correspondence: heqiang@just.edu.cn

**Abstract:** Due to their lightweight, porous and excellent energy absorption characteristics, foam and honeycomb materials have been widely used for filling energy absorbing devices. For further improving the energy absorption performance of the novel tube proposed in our recent work, the nonlinear dynamics software Abaqus was firstly used to establish and verify the simulation model of aluminum-filled tube. Then, the crashworthiness of honeycomb-filled tubes, foam-filled tubes and empty tube under axial load was systematically compared and analyzed. Furthermore, a comparative analysis of the mechanical behavior of filled tubes subjected to bending load was carried out based on the study of dynamic response curve, specific energy absorption and deformation mechanism, the difference in energy absorption performance between them was also revealed. Finally, the most promising filling structure with excellent crashworthiness under lateral load was optimized. The research results show that the novel thin-walled structures filled with foam or honeycomb both show better energy absorption characteristics, with an increase of at least 8.8% in total absorbed energy. At the same time, the mechanical properties of this kind of filled structure are closely related to the filling styles. Foam filling will greatly damage the weight efficiency of the novel thin-walled tube. However, honeycomb filling is beneficial to the improvement of *SEA*, which can be improved by up to 18.2%.

**Keywords:** thin-walled structure; filling structure; numerical simulation; crashworthiness optimization; energy absorption performance

## 1. Introduction

Thin-walled structures have been widely used in automobiles, aviation and other industrial fields, due to their excellent mechanical properties. Therefore, the research on their mechanical properties has always been a hot topic for scholars [1–5]. Galib et al. [6] conducted a comprehensive experimental and numerical study on circular tubes subjected to dynamic load. Zhang et al. [7,8] pointed out that multi-cell square tubes showed better mechanical properties. Alavi Nia et al. [9] conducted axial impact tests on structures with different polygonal cross-sections, and proposed that the multi-cell cross-section was conducive to the improvement of energy absorption performance.

Considering that the traditional thin-walled structure has limited room to improve energy absorption efficiency and stability, it can no longer meet current requirements. Researchers have found that applying biological structural features to structural design can effectively enhance its energy absorption performance [10–13]. Song et al. [14] designed a novel bionic tube with grooves and studied its crashworthiness under lateral impact. The study demonstrated that the innovative design is conducive to improving the energy absorption efficiency of regular structures. Based on the structural characteristics of bamboo, Zou et al. [15] designed a bionic tube and solved its numerical examples under axial/transverse impact. Palombini et al. [16] mechanically explored the special geometry of a

single vascular bundle in bamboo, and the new design proposed has a better improvement in its strength and crashworthiness under dynamic loads. Ferdynus et al. [17] proposed a new type of trigger for the square tube and focused on its effect on the energy absorption indicators achieved (triggering effect).

Metal matrix syntactic foams are high-performance foams consisting of a light-weight matrix and a set of porous fillers. Orbulov and Szlancsik et al. [18–20] carried out a lot of experimental work to characterize its structure–mechanical property relationship. Fiedler et al. [21] analyzed the mechanical properties of the foam with gradient characteristics. Rabiei et al. [22,23] manufactured steel composite metal foam core sandwich panels and studied their quasi-static mechanical properties. These studies show that metal foam has superior mechanical properties and can be used as energy absorption materials.

In order to further enhance the crashworthiness, some scholars fill the regular tubes with lightweight porous materials such as metal foam and honeycomb [24–28]. Li et al. [29,30] conducted bending experiments on foam-filled tubes with different structures. Qi et al. [31] conducted a numerical analyzed mechanical behavior investigation of empty and foam-filled hybrid beams, and optimized their design. Pandarkar et al. [32] elaborated on foam-filled pipes, and the main conclusion was that filling thin-walled structures can improve the stability of the structure. Cakıroglu [33] focused on the quasi-static mechanical properties of honeycomb-filled round pipes and optimized their crashworthiness design. Inspired by biology, Nian et al. [34] proposed a new type of gradient honeycomb-filled round tube and systematically studied its crashworthiness under lateral load. Yao et al. [35] mainly analyzed the dynamic responses of honeycomb-filled structure under various conditions.

In summary, the novel thin-walled tube obtained by filling with foam and honeycomb material has better crashworthiness. Although there are a large number of studies on the foam or honeycomb filling structures, comparative studies of these two filling methods are rarely reported. Therefore, it is important to conduct in-depth research of the mechanical behavior of the novel thin-walled tubes filled with foam and honeycomb, and then to understand the collision behavior and energy absorption characteristics between them more thoroughly. The difference in the dynamic response of the foam and honeycomb-filled novel thin-walled tubes under different filling styles is systematically studied, specifically involving the energy absorption characteristics, peak impact force, deformation mode and load displacement characteristics. The optimization design of the most promising filling structure with excellent crashworthiness is further conducted to maximize the specific energy absorption and minimize the peak collision force.

## 2. Numerical Model

### 2.1. Geometric Model of the Filling Structure

Figure 1 gives the geometric model of the filling structure. $R_{inner}$, $R_{outer}$ and $T$ are the radius and wall thickness of the inner and outer round tubes, respectively. The dotted line in the picture is the angle bisector of the angle $\alpha$, and the intersection of two adjacent oblique lines falls on the intersection of the angle bisector and the section line of the outer tube. The sizes of $R_{inner}$, $R_{outer}$, $T$ and $\alpha$ are, respectively, 15 mm, 30 mm, 1 mm and 90°. Figure 2b exhibits the different filling methods of these novel thin-walled structures. Among these seven filling styles, G is a full filling style and A–F are partial filling styles. The filling structure adopts the following naming rules: F and H indicate foam and honeycomb, and the second letter indicates the filling style. For example: FA means using foam to fill in A style of filling, HA uses honeycomb to fill in style A.

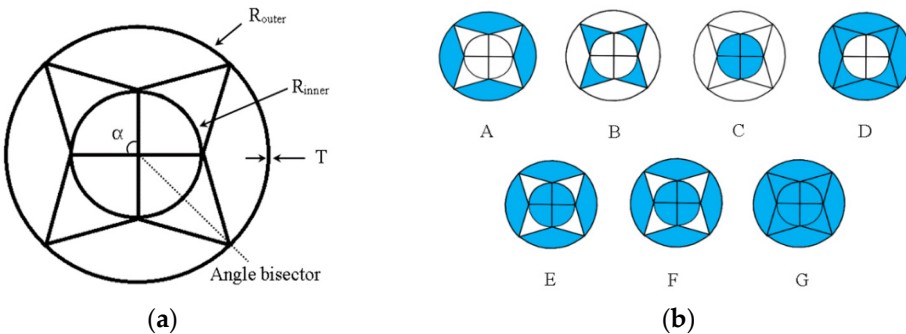

**Figure 1.** Geometric configuration of the filling structure: (**a**) novel thin-walled structure; (**b**) filling styles.

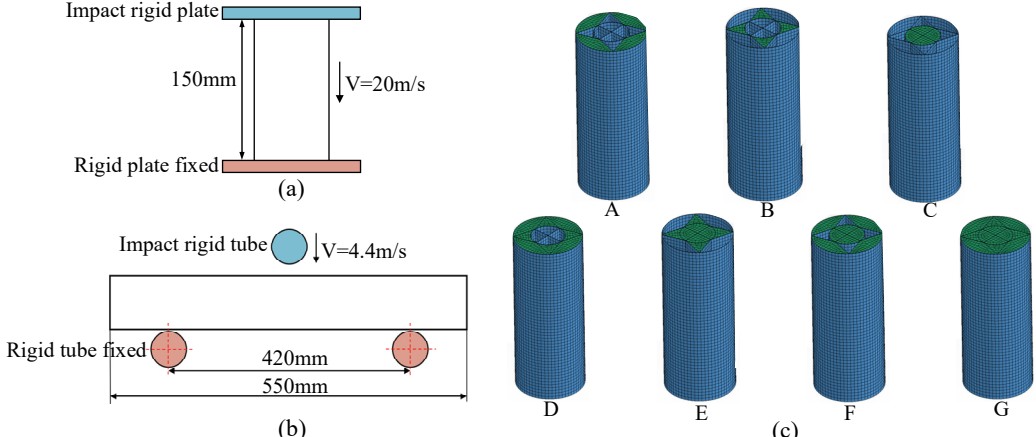

**Figure 2.** Finite element model of the filled structure. (**a**) the calculation model of the axial compression; (**b**) the calculation model of the axial compression; (**c**) Finite element mesh model of thin-walled tube.

### 2.2. Finite Element Model

To study the impact behavior of the infill structure under impact load, a model of the infill structure was constructed by the nonlinear dynamic finite element method for simulation. Figure 2 is the numerical simulation model of infill structure. Part a in Figure 2 is the calculation model of the axial compression of the filling structure. The tube wall and honeycomb are treated as shell, the foam is treated as solid. The impact block and rigid wall are set as rigid bodies. When the impact block impacts the thin-walled tube axially at a speed of 20 m/s, the rigid wall at the bottom is fixed. The contact settings in the dynamic compression process are as follows: the thin-walled tube adopts automatic single-surface contact and the contact between thin-walled tube and the rigid wall is set as surface-to-surface contact. The sensitivity analysis of the mesh shows that the mesh size of the element of 1.5 mm × 1.5 mm is sufficient to produce reliable results. In these contacts, the dynamic and static friction coefficients are set to 0.2 and 0.3. The right side of part a in Figure 2 is the model of the filling structure under different filling methods, with a length of 150 mm.

Part b is the finite element model of the filling structure under lateral load. The indenter and the supports are treated as rigid bodies. The indenter impacts the thin-walled structure vertically downward at 4.4 m/s and the supports are fixed during the impact. The settings of contact properties, mesh size and thin-walled tube length are the same as part a.

### 2.3. Material Properties

The material of the new thin-walled tube and honeycomb filler is aluminum alloy AlMgSi0.5F22 with density $\rho = 2.7 \times 10^3$ kg/m$^3$, elastic modulus $E = 68.566$ GPa, Poisson's

ratio $\nu = 0.29$, yield stress $\sigma_y = 231$ Mpa and ultimate stress $\sigma_{ult} = 254$ Mpa. Considering that aluminum is not sensitive to strain rate, material strain-rate effect can be ignored during simulation analysis [36]. The foam filling is made of foamed aluminum. In order to save calculation cost and ensure sufficient calculation accuracy, crushable foam is used for modeling. The platform stress of lightweight porous materials is very important for its energy absorption. The calculation formula of foam aluminum platform stress $\sigma_p$ [31,37] is as follows:

$$\sigma_p = C_{pow}\left(\frac{\rho_f}{\rho_0}\right)^n \tag{1}$$

In the formula, $\rho_f$ and $\rho_0$ are the density of the foam and foam substrate, respectively. The density of aluminum is $\rho_0 = 2.7 \times 10^3$ kg/m$^3$. $C_{pow}$ and $n$ are constants. According to the test results in literature [38], $C_{pow} = 526$ Mpa and $n = 2.17$. The simplified functional relationship of the foam stress–strain curve is used for simulation, as shown in Table 1 [39]. The Young's modulus of the foam is $E = 64.8$ GPa, the tensile stress cut-off value is 1.11, the rate-sensitive damping is 0.05 and the Poisson's ratio is 0.01 [31].

**Table 1.** Simplified stress–strain relationship of aluminum foam.

| Strain | 0 | $\sigma_p/E$ | 0.6 | 0.7 | 0.75 | 0.8 |
|--------|---|--------------|-----|-----|------|-----|
| Stress | 0 | $\sigma_p$ | $\sigma_p$ | 1.35 $\sigma_p$ | 5 $\sigma_p$ | 0.05 E |

*2.4. Evaluation Index*

Generally, energy absorption (*EA*), average crushing force (*MCF*), maximum collision force (*MIF*), specific energy absorption (*SEA*) and crushing force efficiency (*CLE*) are commonly used evaluation indicators. As a key indicator, specific energy absorption (*SEA*) is often used to evaluate the mechanical performance of thin-walled structures. A larger *SEA* means better crashworthiness. The calculation formula is as follows:

$$SEA = \frac{EA}{M} \tag{2}$$

Among them, *M* is the total mass and *EA* indicates the total absorbed energy of the structure. The calculation formula of *EA* is as follows:

$$EA = \int_0^s F(x)dx \tag{3}$$

In the formula, *S* indicates the displacement of impact force and *F(x)* represents the instantaneous collision force.

*CLE* is an index for evaluating the uniformity and consistency of the collision force. It is another very important evaluation index for crashworthiness. It can be calculated as:

$$CLE = \frac{MCF}{PCF} \times 100\% \tag{4}$$

Among them, *MCF* is the average crushing force, *PCF* is the maximum collision load and the calculation formula of *MCF* is as follows:

$$MCF = \frac{1}{s} \int_0^s F(x)dx \tag{5}$$

*2.5. Validation of the FE Model*

To verify the effectiveness of the numerical model, the results of the axial compression simulation of foam-filled multi-cell square tube (F01, F40) are compared with the reference results in literature [40]. Figure 3 shows that the simulation values in this paper are consistent with the results in reference [40], which have been verified by the theoretical results. Subsequently, the bending behavior of foam-filled square tube in reference [41]

was simulated. Figure 4 shows the comparison between the test results and the numerical results. Both the force–displacement curve and the deformation mode show a high consistency feature. In summary, the finite element model of axial and lateral impact is sufficiently reliable.

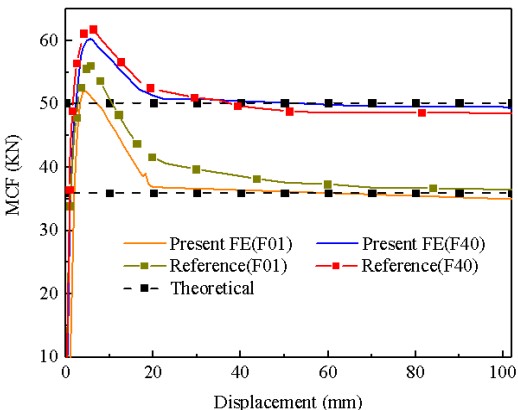

**Figure 3.** The present FE and reference.

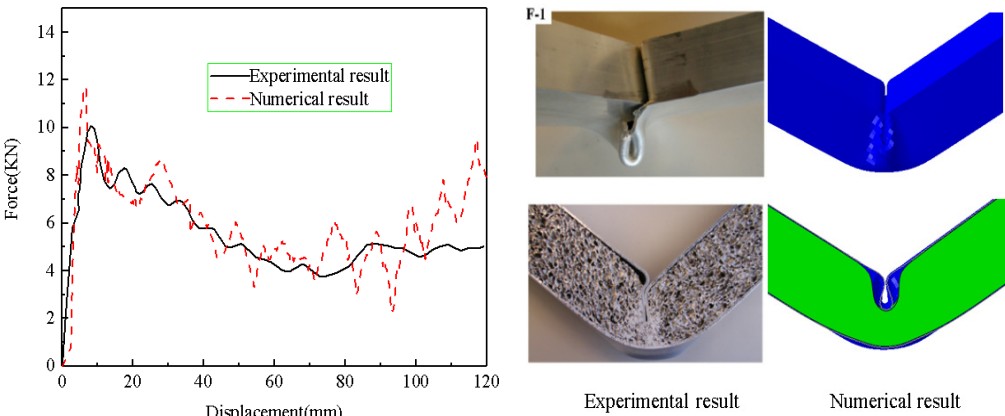

**Figure 4.** Comparison of test results and numerical results.

## 3. Numerical Results

### 3.1. Axial Compression Analysis

In this section, the mechanical behavior of filled structures subjected to axial impact will be studied. The unfilled novel thin-walled tube is also introduced for comparison. Figure 5 shows the *EA* and *SEA* of the filling structure in different filling styles. The *EA* values of the honeycomb- and foam-filled tubes under different filling styles are higher than that of empty tubes (red dotted line) in Figure 5a. In addition, the *EA* of honeycomb/foam-filled tubes shows obvious differences between different filling styles. Partially filled HB/FB has the smallest *EA*, which is, respectively, 8.8% and 17.6% higher than that of empty tube. This shows that whether it is honeycomb or foam filling, the different filling methods of novel thin-walled tube are all conducive to total energy absorption. Additionally, the foam-filling method has better enhancement effect.

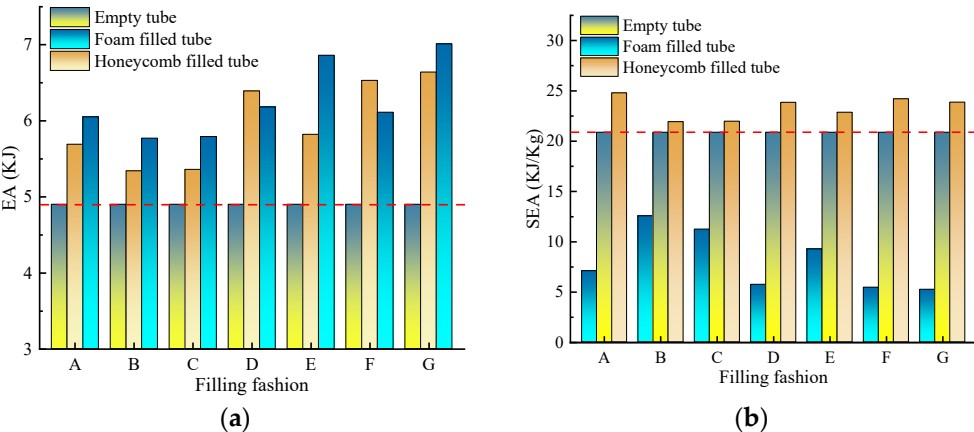

**Figure 5.** *EA* and *SEA* of the filling structure under different filling styles. (**a**) *EA* of filling structure. (**b**) *SEA* of filling structure.

Considering that foam and honeycomb filling may compromise the weight efficiency of the novel structures, Figure 5b compares the *SEA* of the filling structures. As shown in the picture, the *SEA* of the foam-filled tube is relatively low, regardless of the filling method. By contrast, the honeycomb filling method increases the *SEA*. Among the foam/honeycomb-filled tubes, partially filled FB/HA has the largest *SEA*, which is 40.4% lower and 18.2% higher than that of the empty 20.86 KJ tube. Although foam filling plays a positive role in improving mechanical properties, it greatly impairs the weight efficiency. It is worth noting that honeycomb filling just compensates for this defect. Figure 6 shows the peak impact force of the filling structure with different filling methods. The foam-filled structure has the greatest peak force, while the empty tube has the least. This shows that honeycomb is more conducive to reducing peak force than foam filling. Among foam/honeycomb-filled tubes, fully-filled FG/HG has the largest *PCF*, followed by partially filled FE/HD, and partially filled FB/HA is the smallest.

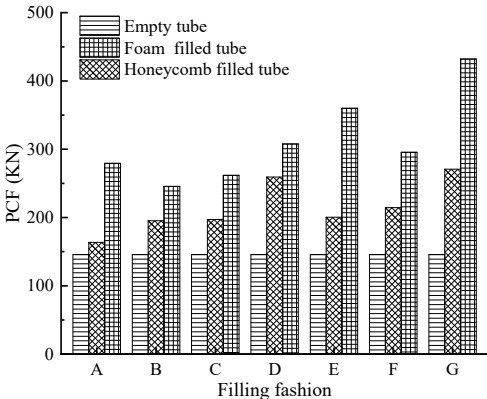

**Figure 6.** *PCF* of filling structure under different filling styles.

From the above analysis, we can see that FB/HA has the best crashworthiness among these novel thin-walled tubes. To have a better understand, Figure 7 exhibits deformation modes of FB, HA and the unfilled thin-walled structure. These three thin-walled structures have all undergone orderly and progressive folding. However, they have different folding characteristics. As shown in the partial enlarged cross-sectional view on the right, the honeycomb-filled tube outside (black solid line ellipse) and the number of internal folds (black solid line box) are larger than foam-filled and empty tubes. Although the number of folds on the outside of the foam-filled tube is the same as that of empty tube, there are more folds on the inside of the foam-filled tube. Meanwhile, the whole structure of the foam-filled structure has a certain degree of plastic deformation (such as the blue solid line

box) when the compression displacement is 60 mm, while more plastic deformation means that more impact energy is absorbed, which explains why the *EA* of FB is greater than that of HA.

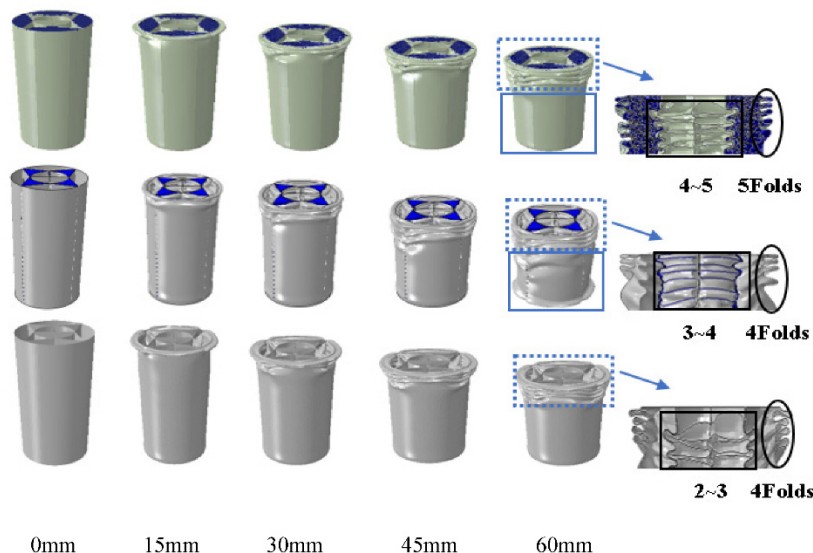

0mm          15mm          30mm          45mm          60mm

**Figure 7.** Deformation mode of filling structure.

### 3.2. Three-Point Bending Analysis

The thin-walled structures will also be subjected to lateral loads in actual use. Thin-walled structures are not only impacted by axial loads, but are sometimes also impacted by lateral loads. Therefore, the bending performance of the structure is very important for its application. Figure 8 shows the *EA*, *SEA*, *PCF* and *CLE* of the filling structure under different filling styles. As shown in Figure 8a, the *EA* of foam- and honeycomb-filled structures are both larger than that of empty tube. Fully filled FG and HG have the largest *EA*, followed by partially filled FA and HD. In particular, under the same filling style, just the *EA* of the honeycomb-filled tube in the filling style C exceeds that of foam-filled tube. This shows that both foam and honeycomb filling will cause the increase in total absorbed energy, and foam filling is more conducive to the growth of *EA* than honeycomb filling. However, it does not represent an increase in its energy absorption efficiency. The *SEA* of honeycomb-filled tube is higher than that of empty tube. The *SEA* of foam/honeycomb-filled pipes showed significant differences. The partially filled FB/HF has the largest *SEA*, which is 27.4%/26.8% lower/higher than the empty tube. The results indicate that the honeycomb filling is an extremely effective means to enhance energy-absorption capacity, and the F filling method may be the best choice.

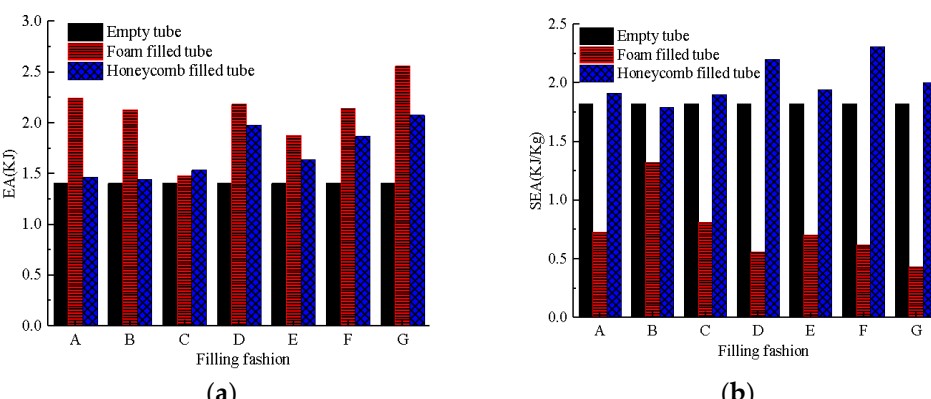

(a)                                                                                          (b)

**Figure 8.** *Cont*.

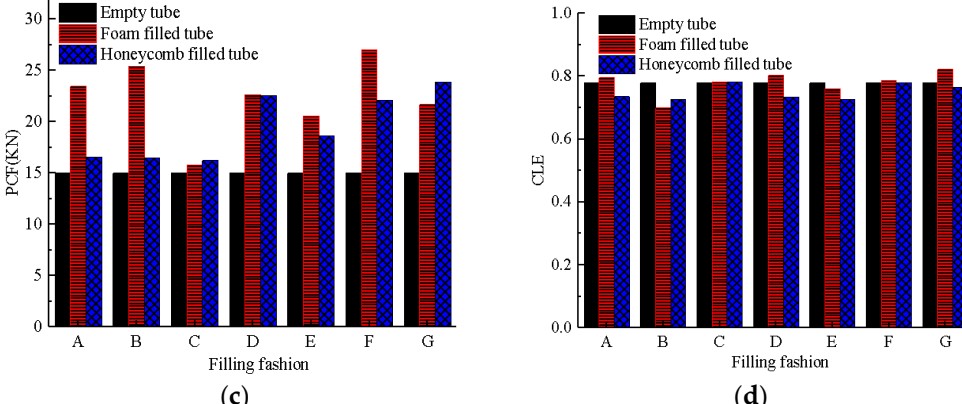

**Figure 8.** The performance indicatorsof the filling structures under different filling methods: (**a**) *EA*; (**b**) *SEA*; (**c**) *PCF*; (**d**) *CLE*.

The *PCF* of the filling structures is also compared in Figure 8c. The *PCF* of foam and honeycomb filling is greater than that of empty tube, and *PCF* of the foam filling tube is the highest. At the same time, the *PCF* of foam and honeycomb tubes are affected by filling method. In Figure 8d, filling techniques do not make *CLE* of the empty tube change too much and the *CLE* did not change significantly with the change of the filling method.

Based on the above, FB/HF has the best energy absorption characteristics among foam/honeycomb-filled tubes. For further understanding the difference in bending performance, the force–displacement curves, specific energy absorption–displacement curves and deformation modes are selected for comparative analysis, as shown in Figure 9. The impact force for the honeycomb filling tube and the empty tube presents the same trend. It first increases sharply and then slowly decreases, while the collision force of the foam-filled tube shows a monotonous increasing trend. The collision force of foam- and honeycomb-filled structures is bigger, and when the loading displacement is equal to 120 mm, the collision force of foam-filled pipes is the largest. This means that foam and honeycomb-filled tubes can withstand a higher level of lateral impact and transmit greater bending moments. In order to better illustrate this point, Figure 9c shows their deformation modes. It can be seen from the map that the partially recessed area of the foam-filled tube is arc-shaped. The effective contact area is larger than that of the honeycomb-filled and empty tube. Although honeycomb filling and empty tube both have the phenomenon of concentrated deformation area, the local recessed area is V-shaped. However, the cross-sectional deformation of the partially recessed area of the honeycomb-filled tube is more obvious than that of empty tube. As shown in the *SEA*-displacement curve in Figure 9b, the *SEA* of honeycomb-filled tube is the largest. This is just the opposite for foam-filled tube. Therefore, foam-filled tubes are not the best choice for crashworthiness.

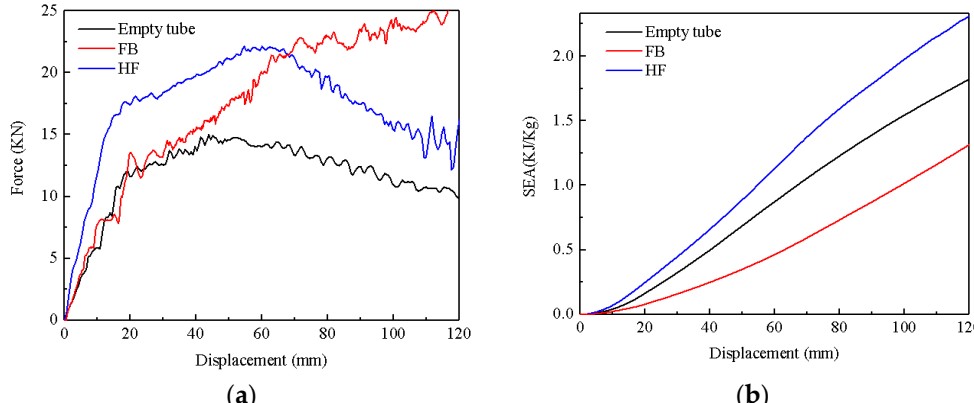

**Figure 9.** *Cont.*

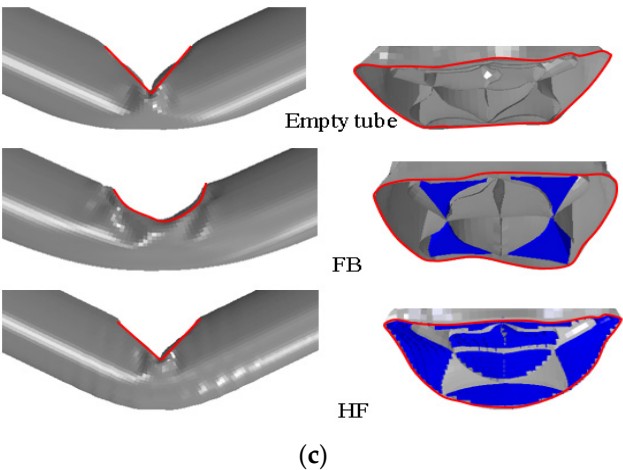

(**c**)

**Figure 9.** Comparison of filled structures: (**a**) force–displacement diagram; (**b**) SEA-displacement diagram; (**c**) deformation mode.

## 4. Multi-Objective Optimization Design

### 4.1. Optimization Problem Set-Up

It is often required that thin-walled structures can absorb the most energy in a certain range of peak stress. Therefore, *SEA* and *PCF* are selected as two objectives of this optimal design. Crashworthiness optimization aims to maximize *SEA* and minimize *PCF*. However, *SEA* and *PCF* are in conflict with each other. Therefore, a multi-objective optimization design is selected to solve this contradictory objective problem [42–45]. From the analysis in Section 3.2, FB and HF have better crash resistance. Moreover, the crashworthiness of HF is better than that of FB. However, the optimal crashworthiness of these structures still depends on different structure and material parameters. Therefore, in this section, the novel thin-walled tube wall thickness $T$, honeycomb wall thickness $t$ and foam density $\rho_f$ are used as design variables. The crashworthiness optimization problem is described as follows:

The optimized expression of FB is as follows:

$$\begin{cases} \text{Minimize } [PCF(T, \rho_f), -SEA(T, \rho_f)] \\ s.t \begin{cases} 0.5 \text{ mm} \leq T \leq 1.5 \text{ mm} \\ 170 \text{ Kg/m}^3 \leq \rho_f \leq 340 \text{ Kg/m}^3 \end{cases} \end{cases} \tag{6}$$

The optimized expression of HF is as follows:

$$\begin{cases} \text{Minimize } [PCF(T, \rho_f), -SEA(T, \rho_f)] \\ s.t \begin{cases} 0.5 \text{ mm} \leq T \leq 1.5 \text{ mm} \\ 0.01 \text{ mm} \leq t \leq 0.1 \text{ mm} \end{cases} \end{cases} \tag{7}$$

The optimized expression of empty tube is as follows:

$$\begin{cases} \text{Minimize } [PCF(T, \rho_f), -SEA(T, \rho_f)] \\ s.t \{0.5 \text{ mm} \leq T \leq 1.5 \text{ mm} \end{cases} \tag{8}$$

### 4.2. Experimental Design

The main experimental design methods include central composite design, Taguchi orthogonal experiment method [45], Latin hypercube design and full-factor experiment. Because the full-factor experiment has good uniformity [10,39], this paper uses the full-factor design to generate 16 sample points (four levels for design variables $T$, $t$, $\rho_f$), as shown in Figure 10. Subsequently, numerical simulation on these sample points is carried out and corresponding response values are obtained.

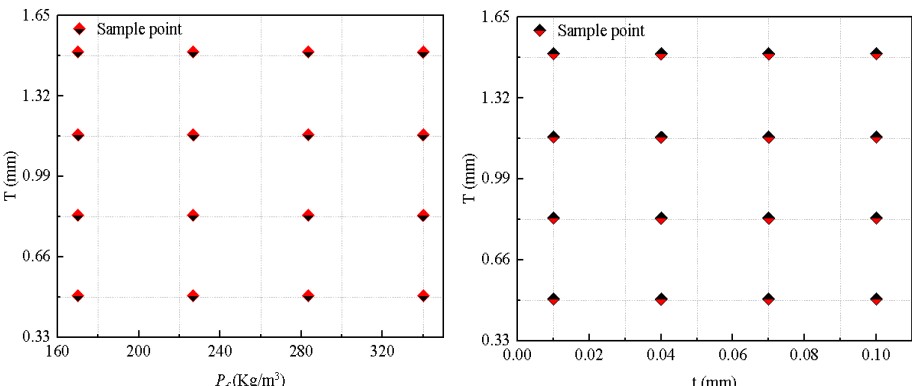

**Figure 10.** Design sample points of FB and HF.

*4.3. Predictive Model*

Since the construction of *SEA* and *PCF* forecasting models is important for the crashworthiness optimization, the accuracy of forecasting models needs to be verified. Firstly, through polynomial regression analysis (PR), the functional relationship between the optimization objective and design parameters is established, and the corresponding prediction model is obtained. Then, the square value of index $R$ ($R^2$), adjusted $R^2$ ($R^2_{adj}$), root mean square error (*RMSE*) and maximum relative error (*MARE*) are used to evaluate the accuracy of this prediction model. The corresponding expression is as follows:

$$R^2 = 1 - \frac{\sum_{i=1}^{n}(y_i - \hat{y}_i)^2}{\sum_{i=1}^{n}(y_i - \hat{y})^2} \tag{9}$$

$$R^2_{adj} = 1 - \left(1 - R^2\right)\frac{n-1}{n-k-1} \tag{10}$$

$$RMSE = \sqrt{\frac{\sum_{i=1}^{n}(y_i - \hat{y}_i)^2}{n}} \tag{11}$$

$$MARE = \max_{i=1,2,\dots n}\left(\frac{|y_i - \hat{y}_i|}{|y_i|}\right) \tag{12}$$

Among them, $y_i$ represents the value of the design points obtained by experiment and numerical analysis, $\hat{y}_i$ is the predicted value of prediction model, $n$ is the number of experimental sample points, $\overline{y}$ represents the average value of $y_i$ and $k$ is the number of non-constant items. Normally, the closer $R^2$ is to 1, the higher the degree of fit; the smaller the *RSME* and *MARE*, the more accurate the prediction model. Table 2 gives the accuracy index of the forecasting model of FB, HF and empty tube. From Table 2, we can see that all $R^2$ values are close to 1, and all *MARE* values are less than 6%. Therefore, it can be considered that these PR mathematical models are accurate enough to be used in crashworthiness optimization.

**Table 2.** Accuracy of the prediction model.

| Objectives | SEA | | | | PCF | | | |
|---|---|---|---|---|---|---|---|---|
| Estimators | $R^2$ | $R^2_{adj}$ | MARE | RMSE | $R^2$ | $R^2_{adj}$ | MARE | RMSE |
| FB | 0.9891 | 0.98 | 2.39% | 0.026 | 0.9949 | 0.9907 | 2.18% | 1.291 |
| HF | 0.9784 | 0.9604 | 2.61% | 0.0307 | 0.9558 | 0.9352 | 5.08% | 0.3869 |
| Empty tube | 0.9987 | 0.9977 | 0.89% | 0.0115 | 0.9981 | 0.9965 | 1.32% | 1.1887 |

*4.4. Particle Swarm Algorithm and Optimization Process*

Since the particle swarm algorithm has the advantages of easy implementation, high accuracy and fast convergence [46,47], this paper uses the particle swarm algorithm to

obtain a Pareto relatively optimal solution of the prediction model. Ten particles are set for tracking and each particle moves 100 times in the constrained space for accuracy. The inertia weight $w$ is an important parameter that affects the pros and cons of the particle swarm algorithm. The solution of the HF prediction model obtained by the particle swarm algorithm under different inertia weights is shown in Figure 11. It can be seen from the picture that when the inertia weight is equal to 0.7, the solutions of the *SEA* and *PCF* prediction models have undergone large oscillations at first, and then stabilized in a certain area. Therefore, when the inertia weight is equal to 0.7, the convergence is best. The specific parameters of the particle swarm algorithm are shown in Table 3. Figure 12 is the flow chart of the optimized design. First, the full-factor experimental design is carried out on the problem of clear optimized design. Further, perform simulation analysis on the sample points to obtain the target response value. Then, the prediction models of *SEA* and *PCF* were constructed through polynomial regression analysis. Finally, the Pareto optimal solution set is given after the calculation of the prediction model by the optimized algorithm.

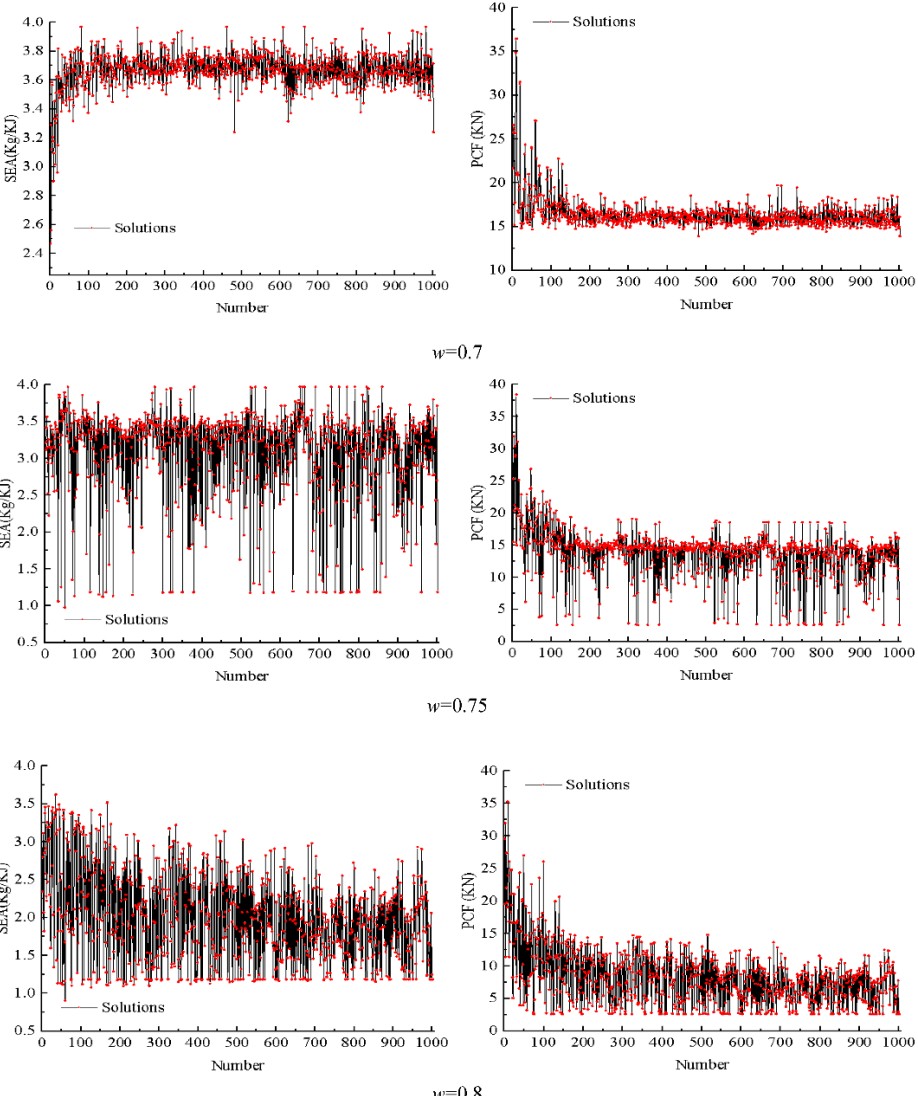

**Figure 11.** The solutions of *SEA* and *PCF* under different inertia weights.

**Table 3.** Parameters of particle swarm algorithm.

| Parameters | Value |
|---|---|
| Number of particles | 10 |
| Maximum number of iterations | 100 |
| Inertia weight | 0.7 |
| Personal learning coefficient | 1.5 |
| Global learning coefficient | 1.5 |

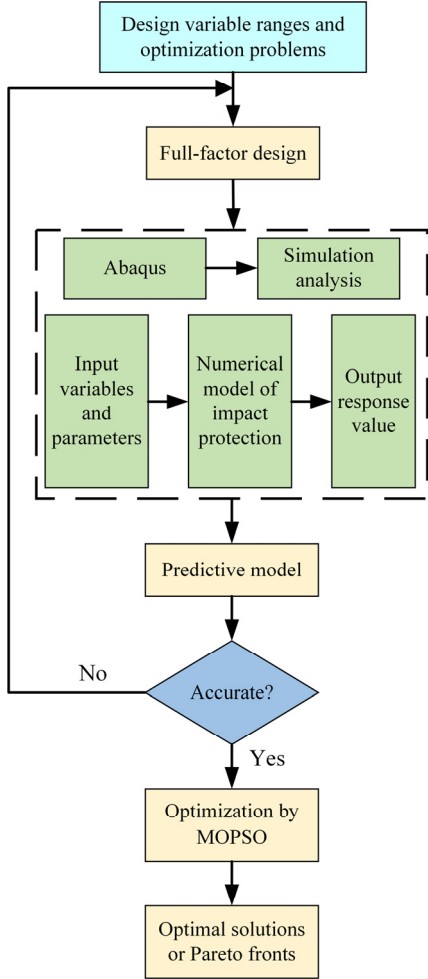

**Figure 12.** Optimization design flow chart.

### 4.5. Multi-Objective Optimization Results

To comprehensively analyze the crashworthiness difference between FB, HF and empty tube more, the Pareto boundary obtained after particle swarm optimization is compared and shown in Figure 13. It can be found from the figure that when the *PCF* is constant, the closer the Pareto boundary is to the left, the greater the specific energy absorption (*SEA*). The Pareto optimal solution set of HF is closest to the left, followed by the empty tube. Therefore, the crashworthiness of HF is better than that of FB and empty tube, and the crashworthiness of FB is the worst. In engineering applications, designers can choose based on requirements of *PCF*. When the *PCF* is less than or equal to 15 KN, the red five-pointed star in the map is the optimal design point of each structure. Figure 14 shows the force–displacement diagrams and deformation modes of these three optimal structures. The *PCF* of the three structures in Figure 14a is less than 15 KN, and HF has the largest *PCF*. In Figure 14b, local recessed area of the HF optimal structure presents an arc shape, while both FB and empty tube present V shapes.

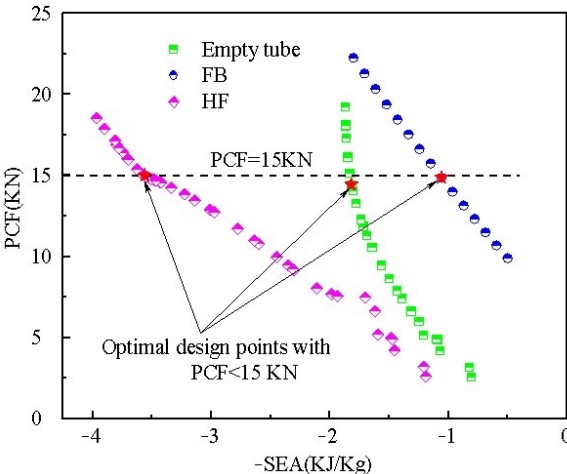

**Figure 13.** Comparison of the Pareto boundary of FB, HF and empty tube.

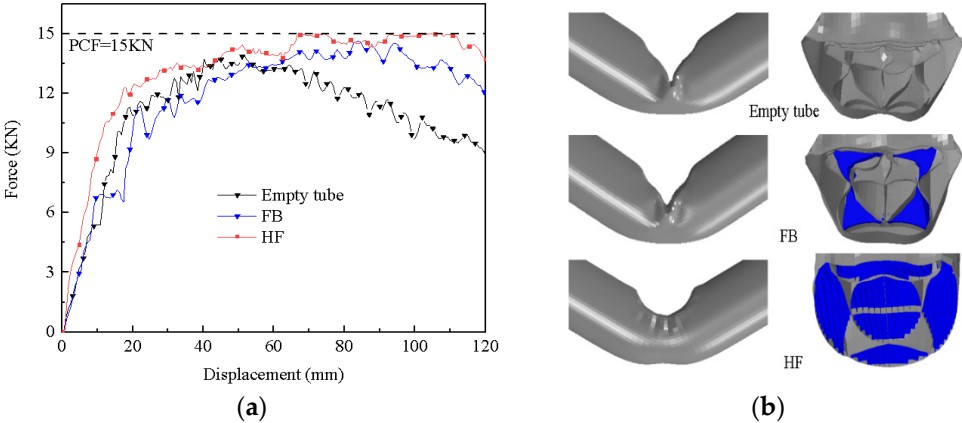

**Figure 14.** Comparison of the optimal structure: (**a**) force–displacement diagram; (**b**) deformation mode.

## 5. Conclusions

To further enhance the crashworthiness of the novel thin-walled structure, foam and honeycomb are used to fill it. The numerical simulation model of the filled structure was first built and verified using the nonlinear dynamic Abaqus software. Then, the impact resistance of honeycomb-filled tubes, foam-filled tubes and empty tube under axial load was systematically compared and analyzed. Furthermore, based on the force–displacement curve, specific energy absorption and deformation model, a comparative analysis of the mechanical behavior of filled tubes subjected to lateral impact was carried out. The optimization design of the most promising filling structure with excellent crashworthiness was finally conducted to maximize the specific energy absorption and minimize the peak collision force. The results of this study show that:

(1) The introduction of honeycomb filling and foam filling enabled the thin-walled structures to absorb more energy. The total absorbed energy increases at least 8.8% compared with the empty tube. At the same time, the crashworthiness of the filling structure was closely related to the filling styles. The foam filling will greatly impair the weight efficiency of the novel thin-walled tube. However, honeycomb filling was beneficial to the improvement of *SEA*, which can be improved by up to 18.2%.

(2) Honeycomb filling was more conducive to the reduction of *PCF* than foam filling under axial load. Among foam/honeycomb-filled tubes, fully filled FG/HG had the largest *PCF*, followed by partially filled FE/HD and FB/HA was the smallest.

(3) Under the action of lateral load, foam- and honeycomb-filled tubes could withstand a higher level of lateral impact and transmit greater bending moments than empty tube.

Among foam/honeycomb-filled tubes, FB/HF was the most promising structure with excellent crashworthiness.

(4) The particle swarm algorithm was further used for crashworthiness optimization design of FB and HF, and the Pareto boundaries were obtained and compared. By way of contrast, the optimal structure of HF showed the best crashworthiness. In practical engineering applications, the use of honeycomb-filled novel thin-walled tube may be the best choice.

**Author Contributions:** Conceptualization, Y.T.; resources, L.L.; software, Y.W.; writing—review and editing, Y.W. and D.X.; methodology, Q.H. All authors have read and agreed to the published version of the manuscript.

**Funding:** This research is supported by The National Natural Science Foundation of China (No. 51705215), The Chinese Postdoctoral Science Foundation (2022M712932) and the Natural Science Fundamental Research Project of Jiangsu Universities (No. 22KJA460003).

**Institutional Review Board Statement:** Not applicable.

**Informed Consent Statement:** Not applicable.

**Data Availability Statement:** Not applicable.

**Conflicts of Interest:** The authors declare there is no conflict of interest regarding the publication of this work.

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
