# Peer review of "Comparative Study and Multi-Objective Crashworthiness Optimization Design of Foam and Honeycomb-Filled Novel Aluminum Thin-Walled Tubes"

_metals, doi:10.3390/met12122163_

Round 1
Reviewer 1 Report
Authors present a well written manuscript, with relevant information and results. The use of simulation for these materials allows their correct design to decrease costs for their manufacturing. Authors used correctly the tools that allow evaluate the behavior of htese materials comparing with experimental results.
Author Response
Point 1: Authors present a well written manuscript, with relevant information and results. The use of simulation for these materials allows their correct design to decrease costs for their manufacturing. Authors used correctly the tools that allow evaluate the behavior of these materials comparing with experimental results.
Response 1: Thanks to the reviewers for their approval of the research and writing of this paper. These days I try my best to do the revising work and dig deeper. I believe that the overall quality of the paper has been improved to a certain extent. I will try my best to analyze the relevant contents if the reviewer points out that there are still some inappropriate modifications.

Reviewer 2 Report
The work is written in a correct manner. The research presented in the work is up-to-date. However, the authors should, address a number of comments to strengthen the work and make it of better quality:
1. in the introduction, it would be appropriate to cite interesting and valuable works on the described research topic: DOI (10.3390/ma13194304)
2. please, at the stage of introduction, specify the clear novelty of the present work by referring to other thematically similar research works.
3. only FEM simulation is presented in the paper there are no experimental results, please let me know if any experimental studies were conducted.
4. please necessarily add to the work a drawing with the FEM model including information about the assigned boundary conditions.
5.The quality of many drawings needs general improvement (Figure 2, 4-7, 12).
6. The conclusions clearly lack a quantitative evaluation of the experimental results.
Author Response
The work is written in a correct manner. The research presented in the work is up-to-date. However, the authors should, address a number of comments to strengthen the work and make it of better quality:
Point 1: In the introduction, it would be appropriate to cite interesting and valuable works on the described research topic: DOI (10.3390/ma13194304)
Response 1: Thanks for the reviewer’s valued suggestion. I felt very sorry that I didn't notice this high-level paper when I was writing my paper. Now I have cited the interesting and valuable works from Ref. [1] in the file “revised-manuscript”.
[1] Ferdynus M, Rozylo P, Rogala M. Energy Absorption Capability of Thin-Walled Prismatic Aluminum Tubes with Spherical Indentations, 2020, 13, 4304.
Point 2: Please, at the stage of introduction, specify the clear novelty of the present work by referring to other thematically similar research works.
Response 2: Thanks for the reviewer’s valued suggestion. The energy absorption capacity of thin-walled tube can be improved through foam or honeycomb filling. Although there are a large number of studies on the thin-walled tubes or metal foam, the comparative study of these two filling methods are rarely reported. The innovation of this article is mainly reflected in the comparative study of the mechanical behavior of the novel thin-walled tube filled with foam and honeycomb. Specifically, we pay attention to the comparison of the collision behavior and energy absorption characteristics between them. Furthermore, the crashworthiness optimization design is carried out based on the better filling method. The relevant content has been tagged in the file “Revised-manuscript”.
Thin-walled tubes have been widely used in many fields, such as automotive, aerospace, engineering, due to its light weight, stable plastic deformation and good energy absorption properties. On the one hand, we have a further understanding of the dynamic behavior of filled tubes subjected to impact load through the study of this article. On the other hand, the results inspire the designers to effectively control the energy absorption of thin-walled tubes by introduction of honeycomb or foam filling. The research method of this paper could serve as reference and guidance for the study of other novel aluminum thin-walled tubes.
All the above is the motivation and novelty of this research. Actually, I have already mentioned part of these messages in the introduction section. After all, the time of review period is limited, and it is impossible for a reviewer to catch every message soon. I will try my best to analyze the relevant contents if the reviewer is still in doubt. However, according to the reviewer’s valued suggestion, I have added some new contents in the “Abstract” and “Conclusion” to better illustrate the innovation and the application of this research.
Point 3: Only FEM simulation is presented in the paper there are no experimental results, please let me know if any experimental studies were conducted.
Response 3: Considering that the experimental data on the mechanical properties of the novel thin-walled tube filled with foam and honeycomb is scarce relatively. The feasibility of the simulation model is hard to be directly proved. This presented me with a dilemma until I studied a large number of existing papers. Actually, numerical simulation is playing an important role in the area of studying the mechanical characteristics of thin-wall metal components under impact, especially in the study of new structures. Many articles just focus on numerical simulation, such as Refs. [1-15]. The composition of this paper mainly refers the related high-quality academic papers.
However, there are some literatures that have analyzed the axial compression simulation of foam-filled multi-cell square tube and the bending behavior of foam-filled square tube [16, 17]. In order to verify the reliability of the finite element model, the results of the simulation model in this paper are compared with the reference results in literature [16, 17]. Both the force-displacement curve and the deformation mode show high consistency feature. In summary, the finite element model of axial and lateral impact is sufficiently reliable. The associated content is detailed in Section 2.5.
Luckily, my works of the mechanical properties of thin-walled tubes, including this manuscript and the other manuscripts (under writing), have gained enough support. Our team had successfully applied for the “National Natural Science Fund”. Part of the relevant test work is being prepared to further study the mechanical properties of thin-walled metallic tubes with aluminum foam filler. However, the test results can not be got in the near future considering the test which includes the setting up of test platform and preparation of specimen will need time for getting ready. I look forward to receiving your understand.
[1] Zhang XC, Liu Y, Wang B, Zhang ZM. Effects of defects on the in-plane dynamic crushing of metal honeycombs. International Journal of Mechanical Sciences 2010;52:1290–98.
[2] Silva MJ, Gibson LJ. The effects of non-periodic microstructure and defects on the compressive strength of two–dimensional cellular solids. International Journal of Mechanical Sciences 1997;39:549–63.
[3] Guo XE, Gibson LJ. Behavior of intact and damaged honeycombs: a finite element study. International Journal of Mechanical Sciences 1999;41:85–105.
[4] Simone AE, Gibson LJ. The effects of cell face curvature and corrugations on the stiffness and strength of metallic foams. Acta Materialia 1998;46:3929–35.
[5] Chen C, Lu TJ, Fleck NA. Effect of imperfections on the yielding of two-dimensional foams. Journal of the Mechanics and Physics of Solids 1999;47:2235–72.
[6] Chung J, Waas AM. Elastic imperfection sensitivity of hexagonally packed circular cell honeycombs. Proceedings of the Royal Society A: Mathematical, Physical and Engineering Sciences 2002;458:2851–68.
[7] Li K, Gao XL, Subhash G. Effects of cell shape and cell wall thickness variations on the elastic properties of two-dimensional cellular solids. International Journal of Solids and Structures 2005;42:1777–95.
[8] Qiao J, Chen C. In-plane crushing of a hierarchical honeycomb. International Journal of Solids & Structures, 2016, 85-6:57-66.
[9] Liu Y, Zhang X C. The influence of cell micro-topology on the in-plane dynamic crushing of honeycombs. International Journal of Impact Engineering, 2009, 36(1):98-109.
[10] Zhu HX, Thorpe SM, Windle AH. The effect of cell irregularity on the high strain compression of 2D Voronoi honeycomb. International Journal of Solids and Structures 2006;46:1061–78.
[11] Li K, Gao XL, Wang J. Dynamic crushing behavior of honeycomb structures with irregular cell shapes and non-uniform cell wall thickness. International Journal of Solids and Structures 2007;44:5003–26.
[12] Wang A, McDowell DL. Effects of defects on in-plane properties of periodic metal honeycombs. International Journal of Mechanical Sciences 2003;45:1799–813.
[13] Chen C, Lu TJ, Fleck NA. Effect of inclusions and holes on the stiffness and strength of honeycombs. International Journal of Mechanical Sciences 2001;43:487–504.
[14] Deqiang Sun, Weihong Zhang, Yucong Zhao. In-plane crushing and energy absorption performance of multi-layer regularly arranged circular honeycombs [J]. Composite Structures 2013; 96:726-735.
[15] Nakamoto H, Adachi T, Araki W. In-plane impact behavior of honeycomb structures randomly filled with rigid inclusions. International Journal of Impact Engineering 2009;36:73–80.
[16] Y Zhang, PZ Ge, MH Lu, XG Lai. Crashworthiness study for multi-cell composite filling structures [J]. International Journal of Crashworthiness. 23 (2018) 32-46.
[17] HR Zarei, M Kroger. Bending behavior of empty and foam-filled beams: Structural optimization [J]. International Journal of Impact Engineering. 35 (2008) 521–529.
Point 4: Please necessarily add to the work a drawing with the FEM model including information about the assigned boundary conditions.
Response 4: Thanks for the reviewer’s valued suggestion. Actually, the finite element model including boundary conditions, cell number, contact property, material properties and so on is built based on the reading and analysis of many articles about the mechanical characteristics of thin-walled structures under impact.
Fig. 2(a) is the calculation model of the axial compression of the filling structure. The impact block moves downwards to crush the tube at velocity of V=20 m/s and the rigid wall at the bottom is fixed. Fig. 2(b) is the finite element model of the filling structure under lateral load. The indenter impacts the thin-walled structure vertically downward at V=4.4m/s and the supports are fixed during the impact. All the above modifications have been marked in the file “revised-manuscript”.
Point 5: The quality of many drawings needs general improvement (Figure 2, 4-7, 12).
Response 5: Thanks for the reviewer’s valued suggestion. These days I try my best to do the revising work and dig deeper according to the reviewers' opinions. At the same time, I try to improve the quality of the drawings mentioned by the reviewer. The description of some pictures is indeed not rigorous enough and will confuse readers. The relevant pictures have been revised in the file “revised-manuscript”. I sincerely look forward to receiving the reviewers’ understanding and I will try my best to analyze the relevant contents if the reviewer points out that there are still some inappropriate modifications.
Point 6: The conclusions clearly lack a quantitative evaluation of the experimental results.
Response 6: Thanks for the reviewer’s valued suggestion. Through the study of this article, we have a further understanding of the dynamic behavior of filled tubes subjected to impact load. On the other hand, the results inspire the designers to effectively control the energy absorption of thin-walled tubes by introduction of honeycomb or foam filling.
Actually, we have gained several useful conclusions which are very qualitative in nature. I'm so sorry that I do not describe it clearly in this paper. For example, the different filling methods of novel thin-walled tube are all conducive to the total energy absorption in Section 3.1. The total absorbed energy increases at least 8.8% compared with the empty tube. In particular, the SEA can be improved by up to 18.2%. In Section 3.2 Three-point bending analysis, the SEA of foam/honeycomb filled pipes showed significant differences. The partially filled FB/HF has the largest SEA, which is 27.4%/26.8% lower/higher than the empty tube. The results indicate that the honeycomb filling is an extremely effective means to enhance energy absorption capacity, and the F filling method may be the best choice.
The above part of contents has been added in the file “revised-manuscript”. As an important part of academic paper, both the 'Abstract' and 'Conclusions' sections should be very qualitative in nature. I have thoroughly examined both of these sections and revised these portions of the manuscript. We refer to other high-level literatures and have quoted some data from analysis results to better illustrate that the energy absorption capacity can be enhanced by the introduction of honeycomb or foam filling. The related content has been noted in both the 'Abstract' and 'Conclusions' sections.

Reviewer 3 Report
The submitted manuscript entitled ‘Comparative study and multi-objective crashworthiness optimization design of foam and honeycomb-filled novel aluminum thin-walled tubes’ deals with the numerical modelling of foam-filled tubes. Basically, the manuscript is interesting and well-written. During the review of the manuscript, two major concerns arose: (i) the physical verification of the numerical results is very limited and restricted to the results of only two (2!) references. Obviously, the professional literature is significantly broader. Please add further comparisons, analyses; (ii) the Authors miss mentioning metal matrix syntactic foams as an effective filler for tubes to enhance the mechanical properties and the energy absorption. Please add references from Szlancsik et al., Rabiei et al., Májlinger et al., Kemény et al., Fiedler et al. etc.
Author Response
The submitted manuscript entitled ‘Comparative study and multi-objective crashworthiness optimization design of foam and honeycomb-filled novel aluminum thin-walled tubes’ deals with the numerical modelling of foam-filled tubes. Basically, the manuscript is interesting and well-written. During the review of the manuscript, two major concerns arose
Point 1: The physical verification of the numerical results is very limited and restricted to the results of only two (2) references. Obviously, the professional literature is significantly broader. Please add further comparisons, analyses;
Response 1: Considering that the experimental data on the mechanical properties of the novel thin-walled tube filled with foam and honeycomb is scarce relatively. The feasibility of the simulation model is hard to be directly proved. This presented me with a dilemma until I studied a large number of existing papers. Actually, numerical simulation is playing an important role in the area of studying the mechanical characteristics of thin-wall metal components under impact, especially in the study of new structures. Many articles just focus on numerical simulation, such as Refs. [1-15]. The composition of this paper mainly refers the related high-quality academic papers.
However, there are some literatures that have analyzed the axial compression simulation of foam-filled multi-cell square tube and the bending behavior of foam-filled square tube [16, 17]. In order to verify the reliability of the finite element model, the results of the simulation model in this paper are compared with the reference results in literature [16, 17]. Both the force-displacement curve and the deformation mode show high consistency feature. In summary, the finite element model of axial and lateral impact is sufficiently reliable. The associated content is detailed in Section 2.5.
Actually, the verification of finite element model is built based on the reading and analysis of many articles about the mechanical characteristics of thin-walled structures under impact. To ensure that the content of model validation is not too redundant, no comparison with other papers has been added at this stage. I sincerely look forward to receiving the reviewer’s understanding and I will try my best to analyze the relevant contents if the reviewer points out that there are still some inappropriate modifications.
[1] Zhang XC, Liu Y, Wang B, Zhang ZM. Effects of defects on the in-plane dynamic crushing of metal honeycombs. International Journal of Mechanical Sciences 2010;52:1290–98.
[2] Silva MJ, Gibson LJ. The effects of non-periodic microstructure and defects on the compressive strength of two–dimensional cellular solids. International Journal of Mechanical Sciences 1997;39:549–63.
[3] Guo XE, Gibson LJ. Behavior of intact and damaged honeycombs: a finite element study. International Journal of Mechanical Sciences 1999;41:85–105.
[4] Simone AE, Gibson LJ. The effects of cell face curvature and corrugations on the stiffness and strength of metallic foams. Acta Materialia 1998;46:3929–35.
[5] Chen C, Lu TJ, Fleck NA. Effect of imperfections on the yielding of two-dimensional foams. Journal of the Mechanics and Physics of Solids 1999;47:2235–72.
[6] Chung J, Waas AM. Elastic imperfection sensitivity of hexagonally packed circular cell honeycombs. Proceedings of the Royal Society A: Mathematical, Physical and Engineering Sciences 2002;458:2851–68.
[7] Li K, Gao XL, Subhash G. Effects of cell shape and cell wall thickness variations on the elastic properties of two-dimensional cellular solids. International Journal of Solids and Structures 2005;42:1777–95.
[8] Qiao J, Chen C. In-plane crushing of a hierarchical honeycomb. International Journal of Solids & Structures, 2016, 85-6:57-66.
[9] Liu Y, Zhang X C. The influence of cell micro-topology on the in-plane dynamic crushing of honeycombs. International Journal of Impact Engineering, 2009, 36(1):98-109.
[10] Zhu HX, Thorpe SM, Windle AH. The effect of cell irregularity on the high strain compression of 2D Voronoi honeycomb. International Journal of Solids and Structures 2006;46:1061–78.
[11] Li K, Gao XL, Wang J. Dynamic crushing behavior of honeycomb structures with irregular cell shapes and non-uniform cell wall thickness. International Journal of Solids and Structures 2007;44:5003–26.
[12] Wang A, McDowell DL. Effects of defects on in-plane properties of periodic metal honeycombs. International Journal of Mechanical Sciences 2003;45:1799–813.
[13] Chen C, Lu TJ, Fleck NA. Effect of inclusions and holes on the stiffness and strength of honeycombs. International Journal of Mechanical Sciences 2001;43:487–504.
[14] Deqiang Sun, Weihong Zhang, Yucong Zhao. In-plane crushing and energy absorption performance of multi-layer regularly arranged circular honeycombs [J]. Composite Structures 2013; 96:726-735.
[15] Nakamoto H, Adachi T, Araki W. In-plane impact behavior of honeycomb structures randomly filled with rigid inclusions. International Journal of Impact Engineering 2009;36:73–80.
[16] Y Zhang, PZ Ge, MH Lu, XG Lai. Crashworthiness study for multi-cell composite filling structures [J]. International Journal of Crashworthiness. 23 (2018) 32-46.
[17] HR Zarei, M Kroger. Bending behavior of empty and foam-filled beams: Structural optimization [J]. International Journal of Impact Engineering. 35 (2008) 521–529.
Point 2: The Authors miss mentioning metal matrix syntactic foams as an effective filler for tubes to enhance the mechanical properties and the energy absorption. Please add references from Szlancsik et al., Rabiei et al., Májlinger et al., Kemény et al., Fiedler et al. etc.
Response 2: Thanks for the reviewer’s valued suggestion. I felt very sorry that I didn't notice these high-level papers [1-6] when I was writing my paper. Now I have added the corresponding contents in the file “revised-manuscript”.
[1] Orbulov I N, Májlinger K. Characteristic compressive properties of hybrid metal matrix syntactic foams. Materials Science & Engineering, A. Structural Materials: Properties, Misrostructure and Processing, 2014, 606, 248-256.
[2] Orbulov I N , Szlancsik A, A Kemény, et al. Compressive mechanical properties of low-cost, aluminium matrix syntactic foams[J]. Composites Part A Applied Science and Manufacturing, 2020, 135:105923.
[3] Szlancsik A , Orbulov I N . Compressive properties of metal matrix syntactic foams in uni- and triaxial compression[J]. Materials Science and Engineering: A, 2021, 827:142081-.
[4] Movahedi N, Vesenjak M, Krstulovi-Opara L, Belova I V, Murch G E, Fiedler T. Dynamic compression of functionally-graded metal syntactic foams. Composite Structures, 2021, 261, 113308.
[5] Marx J , Rabiei A . Study on the Microstructure and Compression of Composite Metal Foam Core Sandwich Panels[J]. Metallurgical and Materials Transactions A, 2020, 51(10):5187-5197.
[6] Marx J , Rabiei A. Tensile properties of composite metal foam and composite metal foam core sandwich panels[J]. Journal of Sandwich Structures and Materials, 2021, 23(8):3773-3793.

Round 2
Reviewer 2 Report
The article has been corrected.